# Cocoa Flavanols Improve Vascular Responses to Acute Mental Stress in Young Healthy Adults

**DOI:** 10.3390/nu13041103

**Published:** 2021-03-27

**Authors:** Rosalind Baynham, Jet J.C.S. Veldhuijzen van Zanten, Paul W. Johns, Quang S. Pham, Catarina Rendeiro

**Affiliations:** 1School of Sport, Exercise and Rehabilitation Sciences, University of Birmingham, Birmingham B15 2TT, UK; RXB585@student.bham.ac.uk (R.B.); J.J.VeldhuijzenvanZant@bham.ac.uk (J.J.V.v.Z.); 2Abbott-Nutrition Division, Research and Development, 3300 Stelzer Road, Columbus, OH 43219, USA; paul.johns@abbott.com (P.W.J.); quangson.pham@abbott.com (Q.S.P.); 3Centre for Human Brain Health, University of Birmingham, Birmingham B15 2TT, UK

**Keywords:** mental stress, cocoa flavanols, endothelial function, forearm blood flow

## Abstract

Mental stress has been shown to induce cardiovascular events, likely due to its negative impact on vascular function. Flavanols, plant-derived polyphenolic compounds, improve endothelial function and blood pressure (BP) in humans, however their effects during stress are not known. This study examined the effects of acute intake of cocoa flavanols on stress-induced changes on vascular function. In a randomised, controlled, double-blind, cross-over intervention study, 30 healthy men ingested a cocoa flavanol beverage (high-flavanol: 150 mg vs. low-flavanol < 4 mg (−)-epicatechin) 1.5 h before an 8-min mental stress task). Forearm blood flow (FBF), BP, and cardiovascular activity were assessed pre- and post-intervention, both at rest and during stress. Endothelial function (brachial flow-mediated dilatation, FMD) and brachial BP were measured before the intervention and 30 and 90 min post-stress. FMD was impaired 30 min post-stress, yet high-flavanol cocoa attenuated this decline and remained significantly higher compared to low-flavanol cocoa at 90 min post-stress. High-flavanol cocoa increased FBF at rest and during stress. Stress-induced cardiovascular and BP responses were similar in both conditions. Flavanols are effective at counteracting mental stress-induced endothelial dysfunction and improving peripheral blood flow during stress. These findings suggest the use of flavanol-rich dietary strategies to protect vascular health during stress.

## 1. Introduction

Stress is highly prevalent in today’s society and has been closely associated with both psychological and physical health. Single episodes of stress have been shown to increase the risk of acute cardiovascular events [1,2,3]. Furthermore, it is well established that stress perturbs cardiovascular activity [4,5], and such responses can contribute to stress-induced cardiovascular events. Indeed, laboratory-based stress tasks have been shown to evoke temporary myocardial ischaemia [6], with peripheral vascular responses during mental stress associated with mental stress-induced ischaemia [7].

Mental stress induces immediate increases in heart rate and blood pressure (BP) in healthy adults and this is accompanied by peripheral arterial vasodilation, as measured by forearm blood flow (FBF) [8,9,10]. In addition to acute responses to stress, studies show that mental stress can lead to post-stress impairments in vascular function [11]. Specifically, transient declines in endothelial function (measured by brachial flow-mediated dilatation; FMD) occur from 15 to 90 min after stress in healthy young adults [12,13,14,15,16], in older adults, postmenopausal women with diabetes and depression, and populations with high cholesterol and metabolic syndrome [17,18,19]. Importantly, reductions in FMD observed following mental stress (1–3% FMD) can be considered significant in the context of future cardiovascular risk, with 1% FMD reduction leading to a 13% increase in cardiovascular risk [20]. Currently, the mechanisms underpinning stress-induced impairments in endothelial function have not been established but studies suggest that reductions in bioavailability of Nitric Oxide (NO), inflammatory markers, and cortisol, as well as increases in vasoconstrictors, such as endothelin-1 (ET-1), may play an important role [21,22,23].

Despite the high prevalence of stress in today’s society, there is limited research investigating the efficacy of strategies to protect the vascular system from the deleterious impact of psychological stress. A 16-week intervention, combining a hypocaloric diet and exercise training in obese children, was shown to reduce BP and increase peripheral vasodilation (as measured by forearm vascular conductance, FVC) during mental stress [24]. This suggests that lifestyle factors, such as diet and exercise, can protect the vasculature from acute mental stress.

Diets rich in fruits and vegetables have been shown to reduce the risk of cardiovascular disease (CVD), myocardial infarction, cardiovascular mortality, and all-cause mortality [25,26]. Flavonoids, a group of small molecules present in most fruits and vegetables, have gained increasing attention in the last 25 years, with epidemiological evidence suggesting significant associations between high intake of flavonoids and a lower CVD-related mortality and morbidity [27,28,29,30]. In particular, flavanols, a sub-group of flavonoids that can be found in cocoa, berries, grapes, apples, and tea, improve human endothelial function (brachial FMD) within 1–3 h of intake [31,32,33,34]. Acute effects of flavanols on endothelial function have been shown to translate into chronic benefits in FMD [35,36,37,38], as well as arteriolar and microvascular vasodilator capacity [37] and BP [36,38,39,40,41]. Circulating flavanol metabolites, particularly (−)-epicatechin-derived [42,43], are believed to drive the beneficial effects of cocoa flavanols on the endothelium by increasing NO bioavailability [34,44,45,46], and reducing ET-1 [47]. Therefore, foods rich in flavanols have the potential to be effective as a dietary strategy to counteract the deleterious impact of mental stress on vascular function. In the current study, we used a randomised, placebo-controlled, double-blind, cross-over intervention design to assess whether the acute intake of cocoa flavanols can improve endothelial function (as measured by brachial FMD) following an episode of mental stress in healthy young adults. We further evaluated the effects of dietary flavanols on peripheral vasodilation (measured by FBF) and BP during mental stress, making this the first study to assess vascular function both during and following mental stress.

## 2. Materials and Methods

### 2.1. Ethical Approval

The present study was approved by the University of Birmingham Ethics Committee (ERN17_1755). Written informed consent was obtained from all participants before enrolment in the study.

### 2.2. Participants

A total of thirty male participants were recruited from the University of Birmingham through email and verbal and poster advertisements. Females were excluded from the study to ensure a more homogenous sample and to minimize the impact of hormonal fluctuations during the menstrual cycle on vascular outcomes. Furthermore, individuals under 18 years old and over 45 years old were also excluded. The other exclusion criteria were: (i) smokers, (ii) consumption of more than 21 units of alcohol per week, (iii) acute illness or infection, (iv) history of cardiovascular, respiratory, metabolic, liver or inflammatory diseases, (v) blood-clotting disorders, (vi) allergies or food intolerances, (vii) weight reducing dietary regiment or dietary supplements, (viii) long-term medication or antibiotics in the prior 3 months. Participants reported to each laboratory session after a 12 h fast, having refrained from alcohol, vigorous exercise, and polyphenol containing foods and drink 24 h before testing (fruits, such as oranges, grapefruit, berries and most vegetables, in particular onions, leeks, peppers, tomatoes, beetroot, as well as cocoa, nuts, olive oil, tea, and coffee). However, foods such as meat, fish, dairy products, bananas, lettuce, pasta and potatoes could be consumed. Whilst polyphenol metabolites may be detected in the blood for up to 48 h [48], most metabolites are excreted within the first 24 h [42] and, as such, we took the practical approach of restricting polyphenols for only 24 h to minimise the burden on our participants. During the study, participants were provided with low-nitrate water to avoid external dietary influences on NO bioavailability [49]. Participants were awarded course credit marks when applicable and a compensation of £25.

### 2.3. Study Design

The study was a randomised, double-blinded, placebo-controlled, cross-over intervention study. Participants completed the acute high and low-flavanol interventions on separate days (at least 7 days apart), starting between 7.30 a.m. and 9.00 a.m. (Figure 1), with the order of the dietary interventions randomised and counterbalanced between participants. Compliance to the food intake requirements were assessed using the 24-h dietary recall. On each visit, participants rested in a supine position for 20 min before pre-intervention baseline measurements were taken: brachial BP, FMD, and FBF, in this order. FBF was measured over an 8 min period, with recordings taken during minutes 2, 4, 6, 8. After pre-intervention baseline assessments, participants consumed either the high-flavanol or low-flavanol drink and 1.5 h later FBF was measured again during an 8 min rest period (Rest) and during an 8 min mental stress task (Stress). Beat-to-beat BP, heart rate (HR), heart rate variability (HRV) and pre-ejection period (PEP) were recorded throughout pre-intervention baseline, rest and mental stress, and data was analysed for the minutes during which FBF was taken. Both FBF and cardiovascular variables were averaged across the 4 min of data collection to provide an overall pre-intervention baseline, rest, and stress value. Brachial BP and brachial FMD were measured 30 (+30) and 90 (+90) min post-stress. Both sessions lasted approximately 5 h, and participants were debriefed after both visits were completed.

### 2.4. Flavanol-Containing Interventions

Cocoa flavanol beverages were prepared by dissolving 8.3 g cocoa powder into 300 mL of commercially available bottled ‘Buxton’ water (containing low levels of nitroso species). The cocoa powders are commercially available (Barry Callebaut, Zurich, Switzerland): the low-flavanol powder was a fat-reduced alkalized cocoa powder (commercial name: 10/12 DDP Royal Dutch) delivering < 4 mg (−)-epicatechin and 4.1 mg of total flavanols per serving; and the high-flavanol cocoa powder was a non-alkalized fat-reduced powder (‘Natural Acticoa’), delivering 150 mg (−)-epicatechin powder and 681.4 mg total flavanols per serving, as used in previous research [50] (Table 1). Both interventions were matched for all other micro and macro-nutrients, including caffeine and theobromine (Table 1). Cocoa powder levels for flavanol monomers, procyanidin and methylxanthines (caffeine, theobromine) were measured by high-performance liquid chromatography (HPLC) as described previously [51,52]. Total levels of polyphenols were also assessed by a Folin-Ciocalteu reagent calorimetric assay as described previously [53]. The dose of flavanol monomers used in the present study is in line with previous studies, shown to be safe and effective in modifying human endothelial function in young healthy adults [33,34,54]. The cocoa powder sachets were labelled with an alphanumeric identifier, and were stored at −20 °C. Intervention beverages were identical in texture, consistency and taste, and were presented in an opaque container with a black opaque straw to ensure double-blindness. The unblinding of the interventions was performed only after all data analyses were completed.

### 2.5. Mental Stress Task

To elicit mental stress the 8 min Paced-Auditory-Serial-Addition-Task (PASAT) was used, previously shown to induce a physiological stress response with good test-retest reliability [9,55,56]. Participants were required to add two sequentially presented single-digit numbers (1–9), by adding the last number they heard to the next number presented. The time intervals between the numbers were reduced every 2 min; from a 2.8 s interval to 2.4 s, 2 s, and 1.6 s in the final 2 min. Participants were filmed and asked to watch themselves on a screen, which they believed was evaluated by 2 independent body language assessors. The participants’ responses were marked by an experimenter, who also sounded a loud aversive noise once every 10 answers; either following an incorrect response or at the end of the 10-number block. The participants were in direct competition with other participants for a £10 voucher and lost points for every incorrect answer. These elements of increasing difficulty, social evaluation, punishment, competition, and reward have been shown to enhance the provocativeness of the task [57]. Immediately after completion of the task, participants were asked to rate how difficult, stressful, competitive, and enjoyable they found the task, and to what extent they were trying to perform well, scored on a 7-point scale ranging from 0 ‘not at all’ to 6 ‘extremely’. Following both visits, participants were informed about the deception of the task.

### 2.6. Cardiovascular Activity

#### 2.6.1. Impedance Cardiography

Heart rate (HR, bpm), heart rate variability (HRV, ms—a measure of parasympathetic activity) and pre-ejection period (PEP, ms—a measure of sympathetic activity) were measured using an electrocardiogram (ECG) and impedance cardiogram (ICG), which was recorded continuously using the Ambulatory Monitoring System, VU-AMS5s (TD-FPP, Vrije Universiteit, Amsterdam, The Netherlands) following published guidelines [58,59]. The VuAMS5fs was connected to 7 Ag/AgCl spot electrodes (Invisatrace, ConMed Corporation; Largo, FL, USA): 3 ECG electrodes (below the right clavicle, between the lower 2 ribs on the right side, and at the apex of the heart on the left lateral margin of the chest), 2 ICG electrodes (at the top end of the sternum at the suprasternal notch and at the bottom of the sternum at the xiphoid process) and 2 ICG electrodes (on the spine, 3 cm above and 3 cm below the upper and lower electrodes, respectively). Offline analyses using VU-DAMS software with manual inspection and correction of ECG and averaged ICG data were used to derive HR, HRV, and PEP. HRV was calculated from the beat-to-beat ECG data as the square root of the mean of the sum of the squared successive differences in cardiac inter-beat intervals. PEP was defined as the time between Q-wave onset and commencement of systole.

#### 2.6.2. Beat-to-Beat Blood Pressure (BP)

Beat-to-beat arterial BP was recorded using a Finometer (Finapres Medical Systems; Amsterdam, The Netherlands), with a small cuff around the intermediate phalanx of the middle finger, during the same periods as FBF (see procedure). Continuous data were recorded via a Power1401 (CED) connected to a computer programmed in Spike2. The recorded output was analysed offline whereby each file was visually inspected, and systolic blood pressure (SBP) and diastolic blood pressure (DBP) were obtained.

#### 2.6.3. Brachial Blood Pressure (BP)

Brachial SBP and DBP were measured using an automated oscillometric blood pressure monitor (Omron HEM-705CP), with a cuff attached to the right upper arm, following at least 10 min rest in the supine position. Three consecutive measurements were taken and only the last two were averaged for each time point.

### 2.7. Forearm Blood Flow

Forearm blood flow (FBF) was measured using venous occlusion plethysmography with a mercury-in-silastic strain gauge. The strain gauge was connected to a plethysmograph (ECG, Hokanson; Jacksonville, WA, USA), producing an output voltage with a frequency of 0–25 Hz. The plethysmograph signal was digitised at 100 Hz with 16-bit resolution, via a Power1401 (CED) connected to a computer programmed in Spike2, conform [9]. One congestion cuff was placed around the wrist (TMC7, Hokanson), inflated for 1 min to supra-systolic blood pressure (>220 mmHg) and another around the brachial region of the upper arm (SC12, Hokanson), inflated for 5 s to above venous pressure (40 mmHg), every 15 s providing 3 blood flow measurements each minute. Calibration and blood flow analysis were undertaken offline using Spike2 (CED). Each increase in limb circumference can be identified as a slope, which were averaged to yield a mean blood flow per minute (Paine et al., 2013). Forearm vascular conductance (FVC) was calculated by dividing FBF by BP per minute of assessment. Due to differences in baseline FBF/FVC between volunteer cohorts collected a few months apart, this data is presented as % change from pre-intervention baseline.

### 2.8. Flow-Mediated Dilatation

Endothelial function of the brachial artery was assessed using flow-mediated dilatation (FMD). A 15–4 Mhz (15L4 Smart MarKᵀᴹ; Terason, MA, USA) transducer was attached to a Terason Duplex Doppler system (Usmart 3300 NexGen Ultrasound; Terason) in combination with an automatic edge-detection and wall-tracking software (Cardio-vascular Suite, Quipu; Pisa, Italy), which allows for continuous measurement of diameter and blood velocity throughout the FMD assessment. Following 20 min of supine rest, the brachial artery was imaged longitudinally at 5–10 cm proximal to the antecubital fossa. Following baseline images for 1 min, a blood pressure cuff placed around the forearm was inflated to 220 mmHg for 5 min to cause ischaemia. This was followed by a rapid deflation for 5 min causing reactive hyperaemia, with continuous image collection for 5 min post-pressure release, in accordance with established guidelines [60,61]. A researcher, blinded to condition allocation and measurement details, analysed all file images. Peak diameter was defined as the largest diameter obtained after occlusion is released. The FMD response was calculated as the relative diastolic diameter change between baseline and peak diameter. Additionally, resting arterial diameter, resting anterograde and retrograde blood flow were also estimated based on a time-average across the first minute of recording.

### 2.9. Statistical Analysis

All data was statistically analysed using the IBM SPSS version 25 software. Cardiovascular and BP dependent variables of interest (HR, HRV, PEP, SBP, DBP) were analysed using a 2-way repeated measures analyses of variance (ANOVA) with dietary intervention (low-flavanol cocoa, high-flavanol cocoa) and time (pre-intervention baseline, rest, stress) as within-subject factors. FBF and FVC were also analysed using a 2-way ANOVA with dietary intervention (low-flavanol cocoa, high-flavanol cocoa) and time (rest % change, stress % change) as main factors. FMD, arterial diameter, anterograde blood flow, retrograde blood flow and resting BP (brachial SBP, brachial DBP) were analysed by 2-way ANOVA with dietary intervention (low-flavanol cocoa, high-flavanol cocoa) and time (pre-intervention baseline, +30, +90) as main factors. Where appropriate, pairwise comparisons using Bonferroni correction were conducted as post-hoc analyses. All values reported in text and tables are mean ± SD and in graphs are mean ± SE. Occasional missing data, due to movement during FBF, machine malfunction of finopress or VU-AMS, are reflected in the reported ‘n’ values. Reactivity scores for all cardiovascular and vascular measures were calculated as the difference between the average stress value and rest. For all analyses, significance was set at *p* < 0.05. Power analyses revealed that a sample size of 30 participants, power at 85% and alpha set at 0.05, allowed the detection of a medium size interaction effect (0.25), for our primary outcome measure, brachial FMD.

## 3. Results

### 3.1. Participant Characteristics

Characteristics of participants are presented in Table 2 (*n* = 30). All participants were young males, aged 19–36, with a healthy body mass index (BMI), HR, BP and FMD, and identified as white European ethnicity. All participants completed the study, and the flavanol beverages were tolerated by all with no adverse events reported.

### 3.2. Mental Stress Task Ratings

Participants perceived the task as equally difficult (*p* > 0.99), stressful (*p* = 0.442), competitive (*p* = 0.310), enjoyable (*p* = 0.630), and tried to perform well (*p* = 0.326) to the same extent after both high and low-flavanol interventions (*n* = 29). Similarly, there was no significant difference in task performance (PASAT score) between interventions (*n* = 29, *p* = 0.713) (Table 3).

### 3.3. Cardiovascular Responses during Acute Mental Stress

Intervention × time ANOVAs revealed an overall time effect for HR (*n* = 13, *p* < 0.001), HRV (*n* = 10, *p* < 0.001) and PEP (*n* = 13, *p* < 0.001) (Figure 2). Post-hoc analysis revealed that HR was significantly higher during mental stress compared to rest (*p* < *0*.001, + 19.88 ± 10.78 bpm and + 15.45 ± 12.42 bpm in low-flavanol cocoa and high-flavanol cocoa, respectively) and pre-intervention baseline (*p* < 0.001). Post-hoc analysis revealed that HRV was significantly lower during mental stress compared to rest (*p* = 0.006, −37.19 ± 38.92 ms and −28.20 ± 22.12 ms in low-flavanol cocoa and high-flavanol cocoa, respectively), and pre-intervention baseline (*p* = 0.004). Post-hoc analysis revealed that PEP was significantly lower during mental stress compared to rest (*p* = 0.002, −26.38 ± 17.90 ms and −23.34 ± 19.72 ms in low-flavanol cocoa and high-flavanol cocoa, respectively) and pre-intervention baseline (*p* = 0.001). There were no significant intervention (HR: *p* = 0.964; HRV: *p* = 0.425; PEP: *p* = 0.833) or intervention × time interaction effects (HR: *p* = 0.483; HRV: *p* = 0.360; PEP: *p* = 0.883).

### 3.4. Blood Pressure during Acute Mental Stress

Intervention × time ANOVAs revealed a significant time effect for SBP (*n* = 30, *p* < 0.001) and DBP (*n* = 30, *p* < 0.001) (Figure 2). Post-hoc analysis showed that SBP significantly increased during mental stress compared to rest (*p* < 0.001, +34.67 ± 23.38 mmHg and +29.84 ± 15.98 mmHg in low-flavanol cocoa and high-flavanol cocoa, respectively) and pre-intervention baseline (*p* < 0.001). Post-hoc analysis revealed that DBP significantly increased during mental stress compared to rest (*p* < 0.001, +19.10 ± 18.34 mmHg and +17.27 ± 9.60 mmHg in low-flavanol cocoa and high-flavanol cocoa, respectively) and pre-intervention baseline (*p* < 0.001). There were no intervention (SBP: *p* = 0.527; DBP: *p* = *0*.876) or intervention × time interaction (SBP: *p* = 0.287; DBP: *p* = 0.716) effects.

### 3.5. Forearm Blood Flow at Rest and during Acute Mental Stress

A significant intervention × time interaction effect was found for FBF (*n* = 29, *p* = 0.025) but not FVC (*n* = 29, *p* = 0.089) (Figure 3). Following high-flavanol cocoa, FBF was significantly higher, compared to low-flavanol cocoa, both at rest (*p* < 0.001, 0.34 ± 0.83% compared to −0.38 ± 0.28%) and during stress (*p* = 0.002, 1.46 ± 1.81% compared to 0.26 ± 0.66%). A significant time effect revealed mental stress to significantly increase FBF (*p* < *0*.001, +0.63 ± 0.51% and +1.12 ± 1.16% in low-flavanol cocoa and high-flavanol cocoa, respectively) and FVC (*p* < 0.001, +0.32 ± 0.32% and +0.51 ± 0.67% in low-flavanol cocoa and high-flavanol cocoa, respectively), compared to rest. A significant intervention main effect showed a significantly greater FBF (*p* = 0.001) and FVC (*p* = 0.001) following high-flavanol cocoa compared to low-flavanol cocoa.

### 3.6. Flow-Mediated Dilatation Following Mental Stress

Brachial FMD, arterial diameter, retrograde and anterograde blood flow are presented at pre-intervention baseline and following stress (+30, +90) in Table 4 (*n* = 30). There was a significant intervention effect (*p* = 0.001) and intervention × time interaction effect (*p* < 0.001), yet no time effect (*p* = 0.146) on brachial FMD. A significant decline in FMD was observed following the low-flavanol cocoa at 30 min (*p* < 0.001, 3.86 ± 1.97% compared to 5.23 ± 1.39%) but not 90 min (*p* = 0.213) post-stress. FMD was significantly higher following high-flavanol cocoa compared to low-flavanol cocoa, at both 30 min (*p* < 0.001, 5.75 ± 1.74% compared to 3.86 ± 1.97%) and 90 min (*p* = 0.026, 5.38 ± 1.50% compared to 4.66 ± 2.22%) post-stress (Figure 4). A significant time effect was revealed in arterial diameter (*p* < 0.001) with a significantly greater arterial diameter at 30 min (*p* = 0.006, +0.07 ± 0.20 mm and +0.30 ± 1.02 mm in low-flavanol cocoa and high-flavanol cocoa respectively) and 90 min (*p* = 0.001, +0.11 ± 0.20 mm and +0.33 ± 1.09 in low-flavanol cocoa and high-flavanol cocoa, respectively) post-stress compared to pre-intervention baseline, with no effects of intervention (*p* = 0.113) or intervention by time interaction (*p* = 0.441). A significant time effect (*p* = 0.046) was shown in retrograde blood flow, with a significantly greater retrograde blood flow observed at 30 min post-stress compared to 90 min post-stress (*p* = 0.032, +7.69 ± 25.72 cm^3^/min and +7.95 ± 15.10 cm^3^/min in low-flavanol cocoa and high-flavanol cocoa respectively), but no significant time changes were detected for anterograde blood flow (*p* = 0.764). There were no effects of intervention in anterograde (*p* = 0.756) or retrograde blood flow (*p* = 0.874), nor intervention × time in anterograde (*p* = 0.762) or retrograde blood flow (*p* = 0.518).

### 3.7. Resting Brachial Blood Pressure Following Mental Stress

Mental stress significantly impacted post-stress SBP (*n* = 30, *p* < 0.001), but not DBP (*n* = 30, *p* = 0.670) (Table 4). Brachial SBP was significantly greater 30 min (*p* < 0.001, +2.93 ± 5.48 mmHg and +4.56 ± 6.90 mmHg in low-flavanol cocoa and high-flavanol cocoa respectively) and 90 min (*p* = 0.002, +2.26 ± 5.72 mmHg and +2.80 ± 4.93 mmHg in low-flavanol cocoa and high-flavanol cocoa respectively) post-stress compared to pre-intervention baseline. No significant intervention (SBP: *p* = 0.923; DBP: *p* = 0.682) or intervention × time interaction (SBP: *p* = 0.554; DBP: *p* = 0.738) effects were detected.

## 4. Discussion

The present study shows that cocoa flavanols prevented the decline in brachial FMD 30 min post-stress, and FMD remained significantly higher following high-flavanol cocoa compared to low-flavanol cocoa 90 min post-stress. Furthermore, high-flavanol cocoa increased FBF at rest and during stress in comparison to low-flavanol cocoa. Perceptions of the stress task and performance on the stress task were not significantly different between conditions, suggesting a consistent stress experience across interventions. Mental-stress induced changes in HR, HRV, PEP, and BP were not affected by the intake of flavanols. To our knowledge, this is the first study to show that plant-derived flavonoids are effective at counteracting mental-stress induced endothelial dysfunction and improving vascular function during mental stress.

Cocoa flavanols were shown to be effective at preventing the decline in brachial FMD experienced 30 min following mental stress. This is in line with previous work, showing benefits of flavanols on endothelial function 2 h post-intake [31,33,34,62,63,64]. Furthermore, the observed decline in FMD (approx. 1.4% FMD) is in agreement with the literature, reporting a 1–3% impairment in endothelial function post-stress in healthy adults [12,13,14,15,16]. Similarly to the current work, one recent study investigated the efficacy of acute intake of Vitamin C (a strong antioxidant) in the context of mental stress, yet reported no effects on endothelial function [21]. However, in contrast to the current findings, their stress task did not induce significant impairments in brachial FMD (at 30 and 90 min post-stress), making the interpretation of this data difficult. Therefore, to our knowledge, this is the first study to show that one dose of dietary flavanols can effectively eliminate post-stress impairments in endothelial function.

As expected, the physiological responses to stress were indicative of both sympathetic activation and parasympathetic withdrawal [8,9]. Whilst flavanols did not influence HR, HRV, PEP, and BP during stress, they led to increased peripheral vasodilation. This confirms previous research showing flavanols’ benefits in FBF at rest [37], and adds to the literature by demonstrating that these benefits persist when the vasculature is challenged by stress. Despite a significantly higher vasodilatation during mental stress following high-flavanol intake, no attenuation in the rise of BP during stress was observed. This disconnection between FBF and BP responses during stress has been reported before. For example, inflammation-induced changes in FBF during acute mental stress were also not accompanied by modifications in BP [9]. In addition, acute intake of cocoa flavanols does not typically lead to changes in BP in young healthy adults [33,65,66,67,68], but only in older individuals [37] or after chronic flavanol intake in overweight [62], hypertensive [36,39,69], and other at-risk populations [54]. In line with our data, previous chronic interventions combining a chronic hypocaloric diet and exercise training (16 weeks) in obese children, have similarly been shown to be effective at increasing FBF and FVC at rest and during stress, but no changes in BP and other cardiac parameters were observed [24]. Whilst flavanols did not impact BP in this study, the positive effect on vasodilation during stress-induced increases in BP, suggests a protective effect on the vasculature and perhaps a more efficient way of responding to stress. More importantly, flavanol-induced vasodilation during stress may have clinical relevance as associations between peripheral vasodilatory responses and myocardial ischaemia have been previously reported [7].

In the current study, flavanol intake did not affect measures of sympathetic and parasympathetic activity during stress. While this is an area of limited research, previous human studies show that grape polyphenols supplementation do not induce changes in muscle sympathetic nervous activity, but attenuate increases in HR during mental stress in hypertensive adults [70]. This may suggest that flavanols’ effects on HR may vary depending on the target population. On the other hand, in vivo rodent studies report that acute intake of pure flavanols ((−)-epicatechin, procyanidins) can increase plasma catecholamines, adrenaline and noradrenaline with varying efficacy [71,72], but to what extent such results translate to humans is currently unclear.

The mechanisms by which flavanols affect the vasculature are not yet well established, but there is evidence from human [34,47,73] and cellular models [44,45,46,74,75] to suggest that the flavanol (−)-epicatechin and its metabolites can increase bioavailability of NO by enhancing eNOS activation through calcium-mediated activation of signalling pathways, such as PI3K, Akt and PKA. As such, it is likely that similar mechanisms underlie the beneficial effects of flavanols on vasodilation and endothelial function during and following stress. Indeed, there is strong evidence to suggest that NO contributes to the rise in FBF during mental stress [8], as pharmacologically inhibiting NO synthase (with NG-monomethyl-L-arginine and atropine) during stress attenuates increases in blood flow [76].

Human studies also show that flavanols can down-regulate bioavailability of the vasoconstrictor ET-1 [47,77], and this may also be relevant in the context of vascular responses to stress. There is in vitro evidence from in vitro studies that cortisol and corticotropin-releasing hormone (CRH)-induced increases in ET-1, known to reduce NO bioavailability, may contribute to the impairment in FMD following mental stress [11]. Furthermore, increases in cortisol have been associated with post-stress impairments in FMD [21] and, similarly, inflammatory markers remain elevated following mental stress [78]. Conversely, previous evidence suggests that cocoa flavanols can attenuate IL-6 production, as well as reduce reactive oxygen species (ROS), promoting resilience to social stress in mouse models [79]. Whilst we did not assess circulating levels of inflammatory and vascular biomarkers in the present study, we hypothesise that flavanols may reduce the stress-induced inflammatory and/or cortisol response, resulting in a reduction in ET-1 and increase in NO bioavailability during stress. The molecular mechanisms underpinning the effects of flavanols during stress should be addressed in future studies.

Interestingly, brachial systolic BP, but not diastolic, remained significantly elevated at 30 and 90 min following stress, while flavanols had no effect. Previous studies have reported elevated systolic and diastolic BP up to 20 min following mental stress [12,13,80], with SBP returning to baseline at a slower rate [80]. We have also observed a significant increase in resting arterial diameter at 30 and 90 min post-stress, suggesting that arteries are more dilated following stress, so increases in BP are not driven by vasoconstriction. To the best of our knowledge, this is the first time that SBP is shown to remain elevated for prolonged periods of time after stress has ceased, and this might have further implications for the vascular system. Whilst this may not be problematic in a healthy population, for patients with hypertension or at risk of CVD, prolonged stress-induced increases in SBP are of great significance for vascular health.

### Limitations

One of the main limitations of the present study is the exclusion of females, which makes the applicability of the present results limited to healthy men. Future studies should focus on the impact of flavanols on stress-responses in women, particularly given the evidence suggesting that there are gender differences in vascular responses to stress [81]. Secondly, previous evidence has shown that polyphenol microbial derived metabolites can be detected in the blood for up to 48 h [48], whilst in the present study we only restricted polyphenol intake for 24 h prior to the study visits. More recent evidence also shows a large reduction in urinary polyphenol metabolites from 24 to 48 h [42], indicating that most metabolites are excreted within the first 24 h. Whilst previous studies suggest that flavanols can modulate NO, ET-1 and IL-6, in the present study we have not assessed these biomarkers and cannot conclude that flavanol-induced vascular responses to mental stress are specifically linked to these mechanisms. A more in-depth investigation of mechanisms of action underlying these responses should be the focus of future work, particularly in regard to modulation of cortisol levels post-stress. Although the efficacy of flavanol intake in improving vascular responses to stress demonstrated in the present study is of value, extending this work to populations at higher risk of cardiovascular diseases (e.g., hypertensives) or mental stress (e.g., care workers) should be addressed in the future. For example, flavanols might be more effective at modulating blood pressure responses during mental stress in these populations.

Finally, we have a lower number of volunteers (*n* = 10/13) with a complete set of cardiovascular data, due to a malfunction in the Ambulatory Monitoring System. Despite a lower *n*, we still detected significant changes in these variables during stress (effect sizes of time effect = 0.598–0.751). In contrast, the effect sizes for the intervention and intervention × time interaction effects (effect sizes = 0.0001–0.107) are considerably lower. It is therefore unlikely that the non-significant results between flavanol interventions are due to low sample size.

## 5. Conclusions

This study demonstrated that acute intake of flavanol-rich cocoa can be an effective dietary strategy to attenuate the transient impairment in endothelial function following mental stress and improve peripheral vasodilation during mental stress. While no benefits in reducing blood pressure during or following stress were observed, such benefits may require continued chronic flavanol intake and future research should explore this avenue.

Importantly, the findings reported in this study have relevance for everyday diet, given that the daily flavanols dosage administered could be achieved through the consumption of a variety of foods rich in flavanols, particularly apples, black grapes, blackberries, cherries, raspberries, pears, pulses, green tea, unprocessed cocoa, and red wine [82]. Finally, these findings can have important implications for the future use of dietary strategies containing plant-derived flavanols to protect the vasculature during periods of increased stress or in populations that are more vulnerable to the effects of mental stress.

## Figures and Tables

**Figure 1 nutrients-13-01103-f001:**
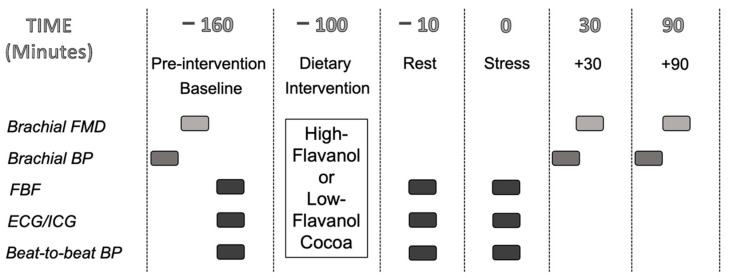
Experimental Study Design.

**Figure 2 nutrients-13-01103-f002:**
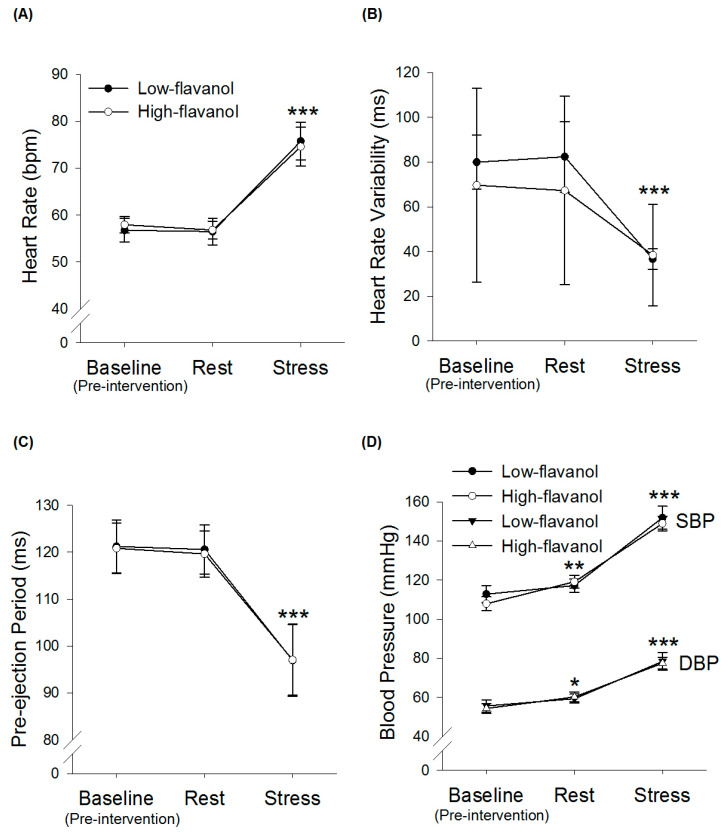
Time course of Heart rate (**A**), Heart rate variability (**B**), Pre-ejection period (**C**) and Systolic and Diastolic blood pressure (SBP, DBP) (**D**) before (pre-intervention baseline) and following either a high-flavanol cocoa or low-flavanol cocoa, at rest and during stress (8-min PASAT). Data are presented as Mean ± SEM. Post-hoc analyses were conducted using Bonferroni’s multiple comparison test. *** HR, HRV, PEP, SBP and DBP were significantly different during stress, compared to rest and pre-intervention baseline, for both conditions (*p* < 0.001). ** SBP was significantly greater during rest compared to pre-intervention baseline (*p* < 0.01). * DBP was significantly greater during rest compared to pre-intervention baseline (*p* < 0.05). (N = 10–13 for HR, HRV and PEP; N = 30 for BP).HR, heart rate; HRV, heart rate variability; PEP, pre-ejection period; SBP, systolic blood pressure; DBP, diastolic blood pressure; BP, blood pressure, PASAT, paced-auditory-serial-addition-task, SEM, standard error of mean

**Figure 3 nutrients-13-01103-f003:**
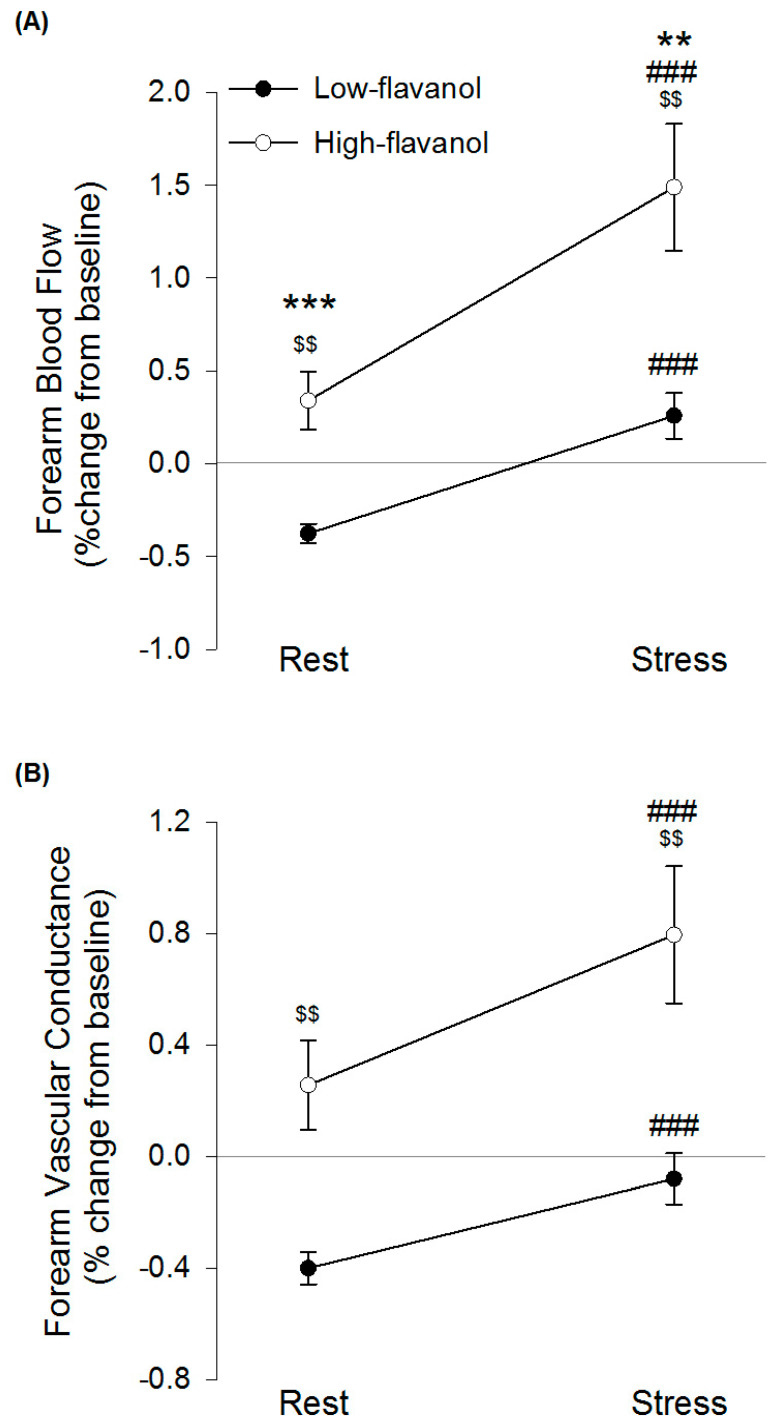
Forearm blood flow (**A**) and Forearm vascular conductance (**B**) following high-flavanol cocoa or low-flavanol cocoa at rest and during stress (8-min PASAT). Values are % change from pre-intervention baseline, and presented as Mean ± SEM. Post-hoc analyses were conducted using Bonferroni’s multiple comparison test. FBF was significantly greater following high-flavanol cocoa compared to low-flavanol cocoa *** during rest (*p* < 0.001) and ** during stress (*p* < 0.01). ### FBF and FVC were significantly greater during stress compared to rest (*p*’s < 0.001). $$ FBF and FVC were significantly greater following high-flavanol cocoa compared to low-flavanol cocoa (*p*’s < 0.01) (N = 29). FBF, forearm blood flow; FVC, forearm vascular conductance; PASAT, paced-auditory-serial-addition-task; SEM, standard error of mean

**Figure 4 nutrients-13-01103-f004:**
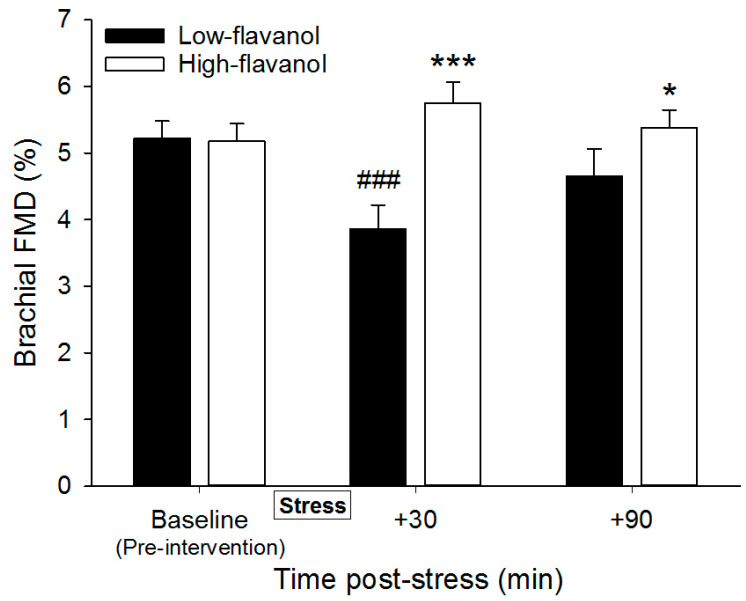
Time course of flow-mediated dilatation of the brachial artery (FMD, %) before and after consumption of either a high-flavanol cocoa or low-flavanol cocoa, at 30 and 90 min post-stress (8-min PASAT). Data are presented as Mean ± SEM. Post-hoc analyses were conducted using Bonferroni’s multiple comparison test. *** FMD was significantly greater at 30 min post-stress (corresponding to 2 h following flavanol consumption) (*p* < 0.001), and * 90 min post-stress (corresponding to 3 h following flavanol consumption) (*p* < 0.05), following high-flavanol cocoa compared to low-flavanol cocoa. ### FMD was significantly lower at 30 min post-stress (corresponding to 2 h following flavanol-consumption), compared to pre-intervention baseline (*p* < 0.001) (N = 30).

**Table 1 nutrients-13-01103-t001:** Composition of cocoa interventions (8.3 g per individual dose) containing high and low flavanol content ^1^.

		High-Flavanol	Low-Flavanol
Total polyphenols	mg	1052.5	143.4
Total flavanols	mg	681.4	4.1
Procyanidins (dimers-decamers)	mg	495.9	ND
(−)-Epicatechin	mg	150.0	<4
(-) and (+)-Catechin	mg	35.5	<4
Theobromine	mg	179.8	179.8
Caffeine	mg	19.5	19.3
Fat	g	1.2	0.9
Carbohydrates	g	4.6	4.4
Protein	g	1.9	1.9
Fibre	g	1.3	2.9
Energy	kcal	19.1	17.0

^1^ As used in previous research [50].

**Table 2 nutrients-13-01103-t002:** Participant baseline characteristics.

Participant Characteristics	Mean ± SD
N	30
Age (years)	23 ± 4.30
BMI (kg/m^2^)	23.66 ± 3.19
HR (bpm)	57.81 ± 7.37
SBP (mmHg)	110.32 ± 20.61
DBP (mmHg)	54.96 ± 14.36
FMD (%)	5.20 ± 1.30

BMI, body mass index; HR, heart rate; SBP, systolic blood pressure; DBP, diastolic blood pressure; FMD, flow-mediated dilatation.

**Table 3 nutrients-13-01103-t003:** Task Ratings and Performance (Mean ± SD, *n* = 29).

Task Ratings (0–6)	Low-Flavanol Cocoa	High-Flavanol Cocoa
Perceived difficulty	4.69 ± 0.89	4.69 ± 0.97
Perceived stressfulness	4.51 ± 1.12	4.66 ± 0.90
Perceived competitiveness	3.90 ± 1.45	3.62 ± 1.57
Perceived enjoyment	2.31 ± 1.28	2.21 ± 1.42
Perception of trying to perform well	5.41 ± 0.68	5.28 ± 0.84
PASAT score (out of 228)	148.80 ± 30.21	147.07 ± 36.13

PASAT, paced-auditory-serial-addition-task.

**Table 4 nutrients-13-01103-t004:** Endothelial function following acute mental stress (Mean ± SD).

Minutes Post-Stress	Low-Flavanol Cocoa			High-Flavanol Cocoa		
	Baseline	+30	+90	Baseline	+30	+90
Brachial SBP (mmHg)	116.93 ± 8.50	119.87 ± 9.65 *	119.19 ± 8.82 *	116.31 ± 9.14	120.87 ± 9.26 *	119.11 ± 10.14 *
Brachial DBP (mmHg)	63.51 ± 8.37	63.70 ± 7.81	63.46 ± 7.39	62.61 ± 7.68	63.76 ± 8.06	63.30 ± 7.66
Arterial diameter (mm)	4.24 ± 0.48	4.30 ± 0.49 *	4.35 ± 0.48 *	4.12 ± 0.40	4.28 ± 0.51 *	4.31 ± 0.56 *
Anterograde Blood flow (cm^3^/min)	150.38 ± 80.66	155.80 ± 82.12	153.89 ± 142.23	152.40 ± 85.80	158.14 ± 117.51	135.11 ± 126.11
Retrograde Blood flow (cm^3^/min)	−24.79 ± 21.99	−29.97 ± 23.98 #	−22.80 ± 26.89	−28.39 ± 28.54	−27.90 ± 25.17 #	−19.95 ± 24.04

* Significantly higher than pre-intervention baseline, regardless of condition *p* < 0.05, # Significantly greater than 90 min post-stress, regardless of condition *p* < 0.05 (*n* = 30).

## Data Availability

The data presented in this study are available on request from the corresponding author.

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
