# Peer review of "Cocoa Flavanols Improve Vascular Responses to Acute Mental Stress in Young Healthy Adults"

_nutrients, 2021, doi:10.3390/nu13041103_

Round 1
Reviewer 1 Report
I was excited to read this interesting study, which examined the effects of acute intake of cocoa flavanols on stress-induced changes in vascular function.
The paper is well written. I would only suggest to authors to conduct sample size and power calculation. Also, HRV with which indices were measured? Can you please provide the readers with the index and its physiological meaning?
Reviewer 2 Report
The manuscript of Baynham and colleagues investigates the effect mediated by cocoa rich in flavanols on vascular responses after an induced mental stress in health men.
General remarks: the manuscript represents one of the first investigating the acute role of flavanols from cocoa on vascular function during and after a mental stress. The topic is of great interest. Minor aspects are listed below.
-Did you fix a priori a range of age for enrolling volunteers? If yes, please specify.
-You stated that subjects were instructed to not consume polyphenol containing foods 24h before the experiments. It is well is well known that polyphenol microbial derived metabolites can be detected in blood up to 48h (e.g. Rodriguez-Mateos, A., Vauzour, D., Krueger, C.G. et al. Bioavailability, bioactivity and impact on health of dietary flavonoids and related compounds: an update. Arch Toxicol 88, 1803–1853 (2014). https://doi.org/10.1007/s00204-014-1330-7), therefore this may represent a limitation of the study.
-Which polyphenol containing foods did you ask to not consume or which foods were allowed before each experiment?
-Did you fix as exclusion criteria unhealthy BMI (e.g. BMI ˃30)? If yes, specify.
-The list of references are not presenting numbers reported in the text.
